# Infrared Thermography as a Diagnostic Tool for the Assessment of Mastitis in Dairy Ruminants

**DOI:** 10.3390/ani14182691

**Published:** 2024-09-16

**Authors:** Vera Korelidou, Panagiotis Simitzis, Theofilos Massouras, Athanasios I. Gelasakis

**Affiliations:** 1Laboratory of Anatomy and Physiology of Farm Animals, Department of Animal Science, School of Animal Biosciences, Agricultural University of Athens (AUA), Iera Odos 75 Str., 11855 Athens, Greece; vkorelidou@aua.gr; 2Laboratory of Animal Breeding and Husbandry, Department of Animal Science, School of Animal Biosciences, Agricultural University of Athens (AUA), Iera Odos 75 Str., 11855 Athens, Greece; pansimitzis@aua.gr; 3Laboratory of Dairy Science and Technology, Department of Food Science and Human Nutrition, Agricultural University of Athens, Iera Odos 75 Str., 11855 Athens, Greece; theomas@aua.gr

**Keywords:** infrared thermography, thermal imaging, udder temperature, mastitis, ruminants, diagnosis

## Abstract

**Simple Summary:**

Infrared thermography is a rapid, contactless technology that can provide real-time temperature measurements, with various applications in livestock science within the framework of precision livestock farming. The aim of the current review paper is to present the updated state of knowledge regarding the detection of mastitis in dairy ruminants using infrared thermography. Therefore, it summarizes the technological advancements, the diagnostic capacity, and the operational and analytical challenges of the method when applied under real-world conditions as a diagnostic tool for mastitis, while factors associated with the performance of the method are critically described and discussed to justify recognized strengths, weaknesses, opportunities, and challenges of the method.

**Abstract:**

Among the health issues of major concern in dairy ruminants, mastitis stands out as being associated with considerable losses in productivity and compromised animal health and welfare. Currently, the available methods for the early detection of mastitis are either inaccurate, requiring further validation, or expensive and labor intensive. Moreover, most of them cannot be applied at the point of care. Infrared thermography (IRT) is a rapid, non-invasive technology that can be used *in situ* to measure udder temperature and identify variations and inconsistencies thereof, serving as a benchmarking tool for the assessment of udders’ physiological and/or health status. Despite the numerous applications in livestock farming, IRT is still underexploited due to the lack of standardized operation procedures and significant gaps regarding the optimum settings of the thermal cameras, which are currently exploited on a case-specific basis. Therefore, the objective of this review paper was twofold: first, to provide the state of knowledge on the applications of IRT for the assessment of udder health status in dairy ruminants, and second, to summarize and discuss the major strengths and weaknesses of IRT application at the point of care, as well as future challenges and opportunities of its extensive adoption for the diagnosis of udder health status and control of mastitis at the animal and herd levels.

## 1. Introduction

The dairy sector contributes to the development of national economies and provides dairy products which are recognized as a valuable source of nutrients [1]. Currently, the growing human population along with the increasing demands for milk and products thereof have resulted in the transformation of the sector towards more intensive farming systems [2]. However, the intensification of production has resulted in emerging health and welfare challenges, with deteriorated udder health status and elevated mastitis cases being among the most significant ones [3,4].

Early detection of mastitis is crucial for its efficient treatment at the animal level and for its prevention at the farm level. Apart from the clinical examination and identification of apparent changes in the udder and milk, several methods and techniques have been developed, particularly for the diagnosis of subclinical mastitis, including (i) the estimation of somatic cell counts (SCCs) in milk, (ii) the identification of biomarkers that are associated with udder health status (enzymes, cytological examination, electrical conductivity), (iii) culturing methods for the isolation and characterization of mastitis-inducing intramammary pathogens, and (iv) imaging techniques (ultrasonography, endoscopy, and infrared thermography) [5,6]. Among them, imaging techniques have the potential to be used *in situ* for the development of novel, rapid, and efficient diagnostic procedures for udder health monitoring [7], facilitating farmers to early detect udder health issues and efficiently manage their herds [8], therefore reducing direct and indirect losses and the costs of repeated treatments [9]. Currently, compared to other imaging techniques, infrared thermography (IRT) has been gaining popularity as a diagnostic tool because it is non-invasive, user friendly, and can be used remotely; despite the promising results, it is still at a developmental stage, with research efforts being currently focused on the overcoming of the limiting factors during its on-farm application, which are associated with the observed inconsistencies of the results [10].

Several studies have been conducted on the application of IRT in detecting mastitis, mainly in cows; however, the most efficient settings and protocols for capturing and analyzing thermal images are yet to be determined. The objective of this paper is twofold: first, to summarize and present the state of knowledge regarding the use of IRT for the assessment of the udder health status in dairy ruminants, describing the most suitable settings and operational aspects, the physiological and environmental factors affecting its performance, and the available recording and analyzing tools of the captured thermograms, and second, to highlight the fundamental strengths and weaknesses of IRT application at the point of care (POC) as well as future challenges and opportunities with regard to (i) its widespread adoption for the diagnosis of udder health status at the animal and herd levels, (ii) user training aspects, (iii) the selection of the most suitable anatomical areas on a case-specific basis, (iv) animal monitoring protocols, and (v) available and developing methods of thermogram analyses.

## 2. Mastitis in Dairy Ruminants

Mastitis is the clinical manifestation of the inflammatory response of the mammary gland to infectious (bacteria, mycoplasma, viruses, and fungi) [11] and non-infectious (disrupted physiological mechanisms, mechanical and thermal injuries, and malformations) agents [12]. It is considered among the major issues challenging dairy ruminants’ health and can lead to substantial production and monetary losses in the dairy industry [13], including the cost of treatment and increased culling and replacement rates, reduced milk production in terms of quantity and quality, impaired animal welfare, and predisposition to other health problems [14,15,16]. It can be categorized as subclinical and clinical according to the absence or presence of clinical signs, respectively, while its severity depends on the age, breed, lactation stage, and health status of the affected animals, as well as the causative agent [7]. Etiological and predisposing factors of mastitis and their interrelationships are complex, with intramammary infections originating either from infected udders or various other environmental elements (e.g., contaminated bedding material, inappropriately disinfected milking machines, filthy hands, and flies) [17]. Bovine mastitis is mainly attributed to bacterial intramammary infections caused by *Enterobacter* spp., *Staphylococcus* spp., or *Streptococcus* spp. [18]. In small ruminants, *Staphylococcus aureus* is the most common etiological factor of clinical mastitis followed by *Streptococcus* spp., *Enterobacter* spp., *Pseudomonas* spp., *Mannheimia* spp., *Corynebacterium* spp., and Lentiviruses. Coagulase-negative staphylococci have been primarily isolated in subclinical mastitis cases [19,20].

## 3. Monitoring of Mastitis

The high prevalence of mastitis, the underdiagnosis of its subclinical form, and the low cure rate urge the development of efficient diagnosis, prevention, and treatment [21]. To address this issue, research efforts need to focus on (i) the proposal of sustainable udder health management protocols, (ii) appropriate hygiene and biosecurity practices, (iii) selective breeding towards genetic resistance against mastitis, and (iv) the development of cost-efficient, reliable, and user-friendly mastitis detection tools with the capacity to be used on field [22,23].

Currently, there are several methods used for the monitoring of mastitis, except for the clinical examination, which is of value mainly in clinical cases. Among these methods, SCC has served as the predominant one for the detection of subclinical mastitis [24]. Based on this method, in the USA, the regulated thresholds for somatic cell count have been set at 1,500,000 cells/mL for goat milk and 750,000 cells/mL for both cow and sheep milk [25]. In Europe, the respective threshold for cow milk has been set to 400,000 cells/mL, while no specific thresholds have been established for sheep and goat milk [26]. The most common method for determining SCC is flow cytometry used by an electro-optical Fossomatic device, which is an automatic cell counter that provides objective and accurate results. To exploit this technology, a remarkable initial investment is needed to purchase the equipment and trained staff to operate it [27]. Another semi-quantitative method that can be used for the estimation of SCC is the California Mastitis Test (CMT); it is a low-cost, rapid, and simple method that can be used in the field without requiring specific expertise by the user; however, it is a subjective semi-quantitative method, with low sensitivity and inconsistent results when it is not applied by the same evaluator [7]. Alternatives include the microscopic method, which is highly precise and regarded as the golden standard for the calibration of relevant technological equipment. This method is time consuming and involves the use of toxic reagents; contrarily, the computer-based method resembling the conventional one is more user friendly and quicker, but expensive [28].

The outcome of the aforementioned methods can be affected by various factors such as the species, breed, stage of lactation, age, parity number, overall udder health status, and milking practices [29]. Also, although they can provide insight into the severity of mastitis, they do not have the capacity to specify the causative agent [30], with bacterial culture and PCR being considered the most appropriate techniques for this scope [19,31]. Although bacterial culture is a low-cost laboratory method for detecting intramammary pathogens that induce mastitis, it is time consuming and demands skilled staff, specialized labs and equipment, and sterile conditions [32]. In addition, bacterial culture can lead to a substantial number of false negative results. Molecular methods, such as PCR, are characterized by high sensitivity and specificity; they provide results within a few hours and have the capacity to identify multiple pathogens (bacteria, yeasts, molds, parasites, viruses) [33]. Nevertheless, they can be costly, unable to differentiate viable from non-viable pathogens, while highly skilled personnel, complex equipment, and sterile conditions are also needed [32].

Through the damage to the epithelial cells’ integrity, mastitis can alter the ionic composition of milk, inducing changes in the milk’s electrical conductivity (Ec) and pH [32]. For this reason, electrical conductivity has been widely exploited and integrated into automated milking systems and milking robots for the in-line automatic detection of mastitis [24]. Although both methods of determining pH and Ec, are cost effective and applicable on field, they display limited diagnostic performance [7], whereas variations in Ec can be attributed to other factors as well (breed, parity, lactation stage, milking interval) [34]. Biomarkers can also be used as indicators of mastitis. The immune system’s reaction to infection and the alterations in blood flow causes the release of enzymes related to inflammation and milk synthesis into milk. Examples of such enzymes are NAGase, serum amyloid A (SAA), haptoglobin (Hp), lactate dehydrogenase (LDH), and plasminogen, which can serve as biomarkers for mastitis [24,27]. Even though these methods are rapid, they cannot be performed on site, and their diagnostic capacity is relatively low compared to molecular methods and microbial culture [32]. The constraints imposed by the aforementioned methods necessitate the development of novel, cost-effective, and accurate tools and tests for the in situ diagnosis of mastitis [30].

## 4. Infrared Thermography

Infrared thermography (IRT), is a real-time, non-invasive, remote, and rapid technology that detects the amount of thermal energy released by an object, generating images, called thermograms, that visualize the temperature distribution [35]. Multiple colors are used to portray distinct temperature levels, with hot areas depicted in white, red, or yellow shades, whereas cold areas are in green and blue (Figure 1) [36]. The underlying principles are based on laws established by Stefan–Boltzmann, Wien, and Planck; as the temperature of a surface exceeds absolute zero, electromagnetic radiation, known as thermal energy, is dissipated in the infrared wavelength (0.75–1000 μm) [37]. This range is further classified as near (0.76–1.5 μm), medium (1.5–5.6 μm), and far infrared (5.6–1000 μm) [38]. Infrared thermography was initially used by military forces to improve night vision capabilities. However, due to its multiple applications, advanced thermal cameras have emerged to be exploited in various fields such as architecture, energy and mechanics, medicine, and agriculture [39]. Its applications in the field of animal science are relatively new; IRT was initially used in equine medicine complementary to ultrasonography and radiography for the diagnosis of limb injuries and disorders and the monitoring of the healing progress [40]. Its effective application in horses contributed to its use in other animal species for the detection of various diseases and inflammatory responses (e.g., mastitis, lameness, parasitic and infectious diseases) [41,42,43,44,45], as well as for other applications, including the evaluation of physiological status and heat stress levels [46,47,48,49], reproduction and estrus detection [50,51,52,53], assessment of nutrition and metabolism [54,55,56,57], and evaluation of animal welfare status and emotions [58,59,60] and their association with the product quality [61]. Despite its wide adoption and promising results, IRT has not become a routine imaging modality yet due to the lack of standardized protocols [62] and the manual obtaining and processing of data from thermograms [63]. Recent advances in machine learning and artificial intelligence show the potential to enhance its automation for inspecting, detecting, and analyzing thermal images and videos, supporting decision-making processes [63,64].

Body temperature is an important indicator of the animal’s metabolism and its overall health and welfare status; hence, it consists of an integral part of clinical examination. The typical method of measuring body temperature via rectal thermometry is time consuming, invasive, and may induce stress on animals that need to be restrained [65]. Moreover, rectal thermometry is used to measure the overall body temperature and does not provide information regarding temperature fluctuations at different parts of the body surface, which can occur as a response to pathological conditions, physiological processes (e.g., during estrous), or other external factors (e.g., ambient temperature, stressful environment, etc.) [66]. On the contrary, thermal imaging can depict body surface temperature distribution, capturing various levels of infrared radiation and providing useful information about inflammation sites or ongoing thermoregulation processes. For example, in the cases of inflammation and heat stress, the observed vasodilatation leads to increased blood flow and hence heat production [67], while on the contrary, in other cases (e.g., cold stress, edema), vasoconstriction leads to decreased blood flow and temperature [68]. Infrared thermography can provide significant insight into animals’ physiological status in a contactless way and, therefore, can be used for inflammation site diagnosis and pain assessment [62]. Nevertheless, the application of thermal imaging is subject to specific limitations induced by various factors. These factors include the operational settings of the thermal camera, as well as environmental, animal-oriented, and operator-derived factors. Thermal cameras’ operational settings refer to emissivity, reflected temperature, humidity, and the camera’s distance from the object [69]. Among these settings, the most essential when employing IRT is emissivity, which is a metric denoting the level of radiation emitted by an item as comparatively assessed to a black body at an equivalent temperature [70]. Emissivity varies according to the surface shape, texture, oxidation, spectral wavelength, temperature, and observation angle [71]. Environmental factors affecting measurements’ accuracy are ambient temperature and humidity, as well as exposure to wind, sunlight, and rain [72]. Animal-oriented factors refer to the animal’s skin attributes, including management practices and animals’ daily routine [72]. Finally, operator-derived sources of variation are also significant and mainly refer to the operator’s skills, familiarization, and training.

## 5. Infrared Thermography for the Evaluation of Udder Health Status

### 5.1. Detection of Mastitis

#### 5.1.1. Cows

Infrared thermography has been extensively employed for the evaluation of udder health status and the detection of mastitis in cows (Table 1 and Figure 2) [73,74,75,76,77,78,79,80,81,82,83,84,85,86,87]. Scott et al. [88] and Barth [89] were among the first researchers to use IRT for the detection of udder inflammation in cattle, with promising results; yet, this review mainly focuses on relevant research over the last 20 years. In a recent study by Machado et al. [84], IRT was found to exhibit high accuracy in detecting subclinical mastitis in cows. In that study, thermograms of the forequarters from a total of 28 cows were captured, and temperatures were assessed in relation to SCC; temperatures derived from the thermal images of front quarters had the highest correlation with SCC (r = 0.87 and r = 0.88 for the left and right forequarters, respectively), while the lowest correlation observed in temperatures of the hind quarters was attributed to their greater exposure to environmental elements (sunlight, wind). Similarly, Velasco-Bolanos et al. [85] evaluated the diagnostic value of IRT in machine- and hand-milked cows with regard to subclinical mastitis and intramammary infections in 105 Holstein cows, as well as the impact of environmental conditions on it; according to their findings, machine milking resulted in an increased udder skin surface temperature (USST) by 1.0 °C, when compared to hand-milking, while the quarters with intramammary infections (IMI) had 1.1 °C increased temperature compared to the uninfected ones. However, no significant associations were observed between USST, the occurrence of subclinical mastitis, and environmental conditions. In another study, Polat et al. [76] examined the interactions between IRT and mastitis indicators (SCC and CMT) and evaluated the capacity of IRT to detect subclinical mastitis (>400,000 cells/mL) compared to CMT. Udder skin surface temperature was found to be significantly correlated with CMT (r = 0.86) and SCC (r = 0.73) in both cases (*p* < 0.001). They also reported that quarters with subclinical mastitis had significantly higher USST, which increased by 2.35 °C compared to the healthy ones; despite the remarkable diagnostic capacity of IRT (sensitivity 95.6% and specificity 93.6%) [76], the authors underpinned the necessity of further replication studies in animals with different phenotypes and under various environments. Moreover, Colak et al. [73] studied the relationship between USST and CMT score in dairy cows (49 Brown Swiss and 45 Holstein Friesian) and concluded that there was a strong positive correlation between them (R^2^ = 0.85, *p* < 0.001); however, they also emphasized on the necessity for further studies to confirm the validity of their findings for different environments and physiological statuses. Hovinen et al. [74] investigated IRT’s capacity to detect clinical mastitis in cows after inducing *Escherichia coli* (*E. coli*) lipopolysaccharide (LPS) in the left hind forequarters of six clinically healthy cows (five Finnish Ayrshire and one Holstein). The authors found that IRT indicated a temperature rise ranging from 1.0 to 1.5 °C in the affected quarters; they also observed that USST increased in tandem with rectal temperature but not with the udder clinical signs, which appeared earlier. Possible explanations for the failure to early detect local signs included increased capillary permeability that caused plasma leakage, resulting in edema and impaired blood circulation, as well as the systemic effects of LPS exposure. In any case, the demand for evaluating IRT diagnostic performance under field conditions in naturally occurring mastitis cases was highlighted in that study. A field study assessing the application of IRT as a primary udder health screening tool was performed by Zaninelli et al. [83] in 155 Holstein Friesian cows derived from three different farming systems. In that study, animals of different lactation periods without clinical signs of mastitis were enrolled; it was found that USST was positively correlated with SCC, with the sensitivity and specificity of the method being 78.6% and 77.9%, respectively, when the threshold was set at 200,000 cells/mL, and 71.4% and 71.6%, respectively, for a threshold equal to 400,000 cells/mL. Metzner et al. [77] inoculated *E. coli* in the teat cistern of the right hind quarter in five healthy Holstein Friesian cows and comparatively assessed temperature distribution in the clinically healthy udder quarters and the infected ones. For these comparisons, various geometric tools were used during the processing of thermograms (polygons, rectangles, lines) along with descriptive parameters (minimum, maximum, range, and mean temperatures); the highest sensitivity and specificity values were achieved using the polygon tool when the maximum temperature was considered. In addition, the highest differences in the median temperature were observed between polygons and rectangles using the maximum temperature (2.06 °C). A year later, the same research team [79] performed a replication study of the developed method by inducing *E. coli* mastitis and examining the changes in USST; both udder skin surface and rectal temperatures initiated increasing 11 h post-inoculation, reaching their peak two hours later and verifying maximum temperature (Tmax) as the most efficient parameter for the detection of mastitis. They also concluded that the time interval between two subsequent examinations should not exceed 2 h. In the study by Pampariene et al. [80], IRT was used to detect udder health-related issues in 152 Lithuanian Black and White cows in ambient temperatures below and above 0 °C and confirmed its accuracy using CMT scores; temperature differences in cases of increased CMT values were 9.6 °C and 4.6 °C higher at temperatures below and above 0 °C, respectively, indicating that IRT could be used instead of CMT under both conditions for the diagnosis of subclinical mastitis. This finding was confirmed by Sathiyabarathi et al. [82], who assessed the efficacy of IRT in detecting subclinical mastitis (SCC > 200,000 cells/mL and Ec > 50 units) in 14 Deoni cows. In that study, quarters affected with subclinical mastitis (37.6 ± 0.29 °C) displayed approximately 1.5 °C and 1.4 °C higher temperature compared to the body temperature (36.1 ± 0.03 °C) and the healthy quarters’ temperatures (36.2 ± 0.06 °C), respectively. In the same study, based on the ROC analysis, an animal was considered to have subclinical mastitis if the temperature difference between the body and the udder skin surface was above 0.58 °C, while a strong correlation (R^2^ > 0.95) between IRT measurements and other indicators of mastitis, such as SCC and Ec, was evidenced; indeed, when the temperature difference exceeded 1.5 °C, the mean milk SCC surpassed 300,000 cells/mL. Satheesan et al. [87] determined specific cut-off temperature points for the udder according to the infection status and correlated udder temperature fluctuations with stress and inflammatory indicators. They reported a 2–3 °C increase in teat apex temperature, teat skin surface temperature (TSST), and USST in Sahiwal cows with subclinical (CMT= 1 and SCC between 2 × 10^5^ and 5 × 10^5^ cells/mL) and clinical mastitis (CMT = 2 and SCC > 5 × 10^5^ cells/mL) attributed to increased blood circulation caused by the inflammatory response. Significant differences in milk pH and conductivity were reported between healthy cows (SCC < 2 × 10^5^ and no previous mastitis) and those with clinical mastitis. In addition, the authors noticed that as the infection progressed, cortisol and acute phase protein (APP) levels increased, while prolactin decreased, emphasizing the importance of developing algorithms to automate the delineation of udder boundaries. Gayathri et al. [86] captured thermal images of the short milking tube of the milking machine (SME), USST, TSST, and eyeball (EBST) to detect mastitis in Sahiwal cows. The study revealed that cows with subclinical (CMT = 1 or 2 and SCC > 2–5 × 10^5^ cells/mL) and clinical (CMT = 3 and SCC > 5 × 10^5^ cells/mL) mastitis exhibited higher USST by 1.6 °C and 2.5 °C at pre-milking, 0.8 °C and 1.3 °C during milking, and 0.7 °C and 1.6 °C at post-milking, respectively, in comparison to healthy cows (CMT = 0 and SCC < 2 × 10^5^ cells/mL). In addition, TSST was higher by 1.1 °C and 0.8 °C in the group with subclinical mastitis and by 1.8 °C and 1.5 °C in the group with clinical mastitis at pre- and post-milking, respectively. The same authors also reported a strong correlation between subclinical mastitis and USST, TSST, EBST, and CMT, with subclinical mastitis increasing by 1.1 °C and 2.0 °C in animals with subclinical and clinical mastitis, respectively, possibly attributed to immunological and inflammatory responses triggered by pathogen invasion. This study suggested that subclinical mastitis can be an effective alternative to USST and TSST for the assessment of mastitis status, eliminating variances in skin or hair color and emissivity.

Except for the studies indicating IRT as a reliable tool for the diagnosis of mastitis, other studies have produced controversial results on this aspect. For example, Bortolami et al. [78] evaluated the accuracy of IRT in detecting subclinical mastitis in 98 Holstein Friesian cows; they also examined its capacity to differentiate subclinical mastitis cases based on the causative pathogens. They found that the average USST was negatively associated with SCC, attributing it to the decreased functionality and blood flow in the udders with subclinical mastitis, while no significant association between specific mastitis-causing bacteria and USST was found. In contrast, uninfected quarters exhibited higher USST. In that study, although USST demonstrated the diagnostic potential of IRT in identifying subclinical mastitis, it was unable to differentiate between pathogens. In the study by Porcionato et al. [75], the merit of IRT in detecting subclinical mastitis (SCC ≥ 200,000 cells/mL and isolation of 1 pathogen) in Gyr cows was investigated. For this purpose, they evaluated USST at three distinct levels based on the height of the udder in the dorsal-ventral direction (upper, median, lower) and examined their relationship with SCC and the bacterial infection status; their findings revealed that the upper area of the udder displayed the highest temperatures, while no correlation was evidenced between the temperatures recorded at different udder levels, SCC values, and the type of the isolated bacteria, concluding that IRT was not an effective tool to diagnose the occurrence of mastitis. The association between SCC and several temperature parameters of the udder was studied by Byrne et al. [81] to propose an automated prediction algorithm for mastitis detection. All the measurements were performed within the captured area enclosed around the ventral face of each quarter excluding teats and background elements (e.g., legs). They established two thresholds at 32.0 °C and 35.0 °C to eliminate the effects of hair or dirt during the analysis and calculated the average temperatures by focusing on pixels within the 0.5 °C and 2.0 °C of the maximum temperature. An insignificant relationship between SCC and USST was found, suggesting that IRT could not be employed for the detection of mastitis.

#### 5.1.2. Buffaloes

Infrared thermography of the udder has been used by Gayathri et al. [90] to detect subclinical and clinical mastitis in Murrah buffaloes throughout different seasons (India). The mean USST and TSST ranges were as follows: (i) 30.3–36.8 °C and 30.5–36.0 °C in healthy buffaloes (CMT = 0), (ii) 32.5–38.6 °C and 32.9–37.6 °C in the subclinical mastitis group (CMT = 1 and 2), and (iii) 34.3–40.0 °C and 34.5–39.1 °C in the clinical mastitis group (CMT = 3). The authors reported a positive correlation between CMT and mean USST and TSST, indicating that animals with mastitis exhibited higher USST and TSST across all months. The highest USST was recorded during the rainy season (July–August), followed by summer (April–June), autumn (September–November), and winter (December–March) for all categories. The authors suggested that IRT can serve as a valuable tool for determining udder health status and underpinned the significance of regular udder and teat temperature surveillance for the establishment of specific cut-off points (Table 1).

#### 5.1.3. Sheep

The available research on the use of IRT for the assessment of udder health in sheep is limited [91,92] (Table 1 and Figure 3). Martins et al. [91] evaluated IRT’s effectiveness in identifying subclinical mastitis cases; they observed that ewes with subclinical mastitis (responsive mammal lymph nodes, mild udder rigidness, mildly positive CMT and SCC between 250,000 and 500,000 cells/mL) demonstrated higher udder temperatures compared to both the healthy ones and the ewes with clinical mastitis (responsive mammal lymph nodes, udder rigidness, positive CMT and SCC > 500,000 cells/mL). They concluded that ewes with clinical mastitis experience severe and persistent inflammation and tissue damage in their udders, thereby losing their functionality and disrupting the tissue integrity, resulting in reduced blood flow and temperature. In another study, Castro-Costa et al. [92] examined the capacity of IRT to detect intramammary infections in Manchega and Lacaune ewes, either naturally occurring or after the infusion of *E. coli* endotoxin. Contrary to the finding by Martins et al. [91], they did not find significant differences in USST between healthy udders and those exhibiting subclinical or clinical mastitis. Also, in that study, breed-related differences in USST were observed, with Manchega ewes displaying lower USST values, likely due to the hairier udders. Moreover, milking was found to affect skin surface temperature at teats, attributable to increased blood flow within teat tissues and hyperemia on the teat wall triggered by repetitive compressions applied to the teat. In addition, the authors reported that USST levels were lower during the morning and before milking, possibly due to the effects of the circadian rhythm, physical activity, and feeding patterns. In particular, early in the morning, body core temperature is lower than in the afternoon, and animals display lower activity resulting in reduced heat production and blood circulation [93]. In addition, feeding is associated with increased blood flow to support digestive organs [94]; hence, animals display lower temperatures early in the morning when they have not been fed.

**Table 1 animals-14-02691-t001:** Studies where infrared thermography was used for the diagnosis of mastitis.

Animals	Time	Temperature (°C) (Mean ± SD/Mean ± SE)		Sensitivity/Specificity/ Accuracy (%)	Reference
		Healthy	Mastitis		
COWS					
49 Brown Swiss and 45 Holstein Friesian cows	BM	33.2 ± 0.52 ^SD^	CMT (i) L1: 34.1 ± 0.29 ^SD^ (ii) L2: 35.0 ± 0.25 ^SD^ (iii) L3: 36.2 ± 0.23 ^SD^	-	[73]
70 Gyr cows	AMM	A. SCC: <10^5^ cells/mL (i) Upper: 34.0 ± 0.02 ^SE^ (ii) Median: 32.7 ± 0.05 ^SE^ (iii) Lower: 32.2 ± 0.06 ^SE^ B. SCC: 10^5^–2 × 10^5^ cells/mL (i) Upper: 34.0 ± 0.04 ^SE^ (ii) Median: 32.4 ± 0.08 ^SE^ (iii) Lower: 32.5 ± 0.09 ^SE^	A. SCC: >2 × 10^5^–3 × 10^5^ cells/mL (i) Upper: 33.7 ± 0.12 ^SE^ (ii) Median: 32.9 ± 0.15 ^SE^ (iii) Lower: 32.0 ± 0.05 ^SE^ B. SCC: >3 × 10^5^ cells/mL (i) Upper: 34.1 ± 0.02 ^SE^ (ii) Median: 32.6 ± 0.05 ^SE^ (iii) Lower: 31.6 ± 0.04 ^SE^	-	[75]
62 Brown Swiss cows	BMM, BAM	33.5 ± 0.09 ^SE^	A. SCC: 35.8 ± 0.08 ^SE^ B. CMT: (i) L1: 34.6 ± 0.12 ^SE^ (ii) L2: 35.7 ± 0.07 ^SE^ (iii) L3: 36.3 ± 0.07 ^SE^	SCC: 4 × 10^5^ cells/mL (Se: 95.6, Sp: 93.6, Ac: 98.5)	[76]
5 Holstein Friesien cows	/2h	34.1–37.7	34.5–39.7	Se: 100.0, Sp: 96.0	[77]
98 Holstein Friesien cows	-	33.1 ± 0.17 ^SD^	(i) *Staphylococcus aureus:* 32.0 ± 0.23 ^SD^ (ii) *Staphylococcus agalactiae:* 32.9 ± 0.36 ^SD^ (iii) *Staphylococcus uberis:* 31.9 ± 0.24 ^SD^ (iv) Coagulase-negative staphylococci: 32.9 ± 0.33 ^SD^	-	[78]
14 Deoni cows	BMM, BAM	(i) Morning: 36.2 ± 0.06 ^SD^ (ii) Evening: 37.1 ± 0.21 ^SD^	37.6 ± 0.29 ^SD^	Se: 54.1–100.0 Sp: 69.2–100.0	[82]
155 Holstein Friesian cows	BM	Tmax (i) SCC: 2 × 10^5^ cells/mL 34.2 ± 0.17 ^SE^ (ii) SCC: 4 × 10^5^ cells/mL 34.4 ± 0.16 ^SE^	Tmax (i) SCC: 2 × 10^5^ cells/mL 35.8 ± 0.15 ^SE^ (ii) SCC: 4 × 10^5^ cells/mL 36.1 ± 0.22 ^SE^	(i) SCC: 2 × 10^5^ cells/mL Se: 78.6, Sp: 77.9 (ii) SCC: 4 × 10^5^ cells/mL Se: 71.4, Sp: 71.6	[83]
105 Holstein cows	During predawn and morning hours	32.4–32.6	(i) SCM: 32.9 (ii) IMI/MG: 33.5	A. Hand milking (i) SCM: (Se: 53.0, Sp: 89.0, Ac: 71.0) (ii) MG/IMI: (Se: 83.0, Sp: 93.0, Ac: 88.0) B. Machine milking (i) SCM: (Se: 42.0, Sp: 97.0, Ac: 70.0) (ii) MG: (Se: 82.0, Sp: 89.0, Ac: 85.0) (iii) IMI: (Se: 82.0, Sp: 98.0, Ac: 90.0)	[85]
25 Sahiwal cows	BMM, AMM, during morning milking	(i) USST BM: 35.0 ± 0.11 ^SE^ Milking: 35.7 ± 0.06 ^SE^ AM: 35.8 ± 0.06 ^SE^ (ii) TSST: BM: 35.0 ± 0.12 ^SE^ AM: 35.5 ± 0.05 ^SE^ (iii) SCM: 32.8 ± 0.12 ^SE^ (iv) EBST: 36.4 ± 0.15 ^SE^	A. SCM (i) USST: BM: 36.6 ± 0.05 ^SE^ Milking: 36.4 ± 0.02 ^SE^ AM: 36.5 ± 0.03 ^SE^ (ii) TSST: BM: 36.1 ± 0.04 ^SE^ AM: 36.3 ± 0.02 ^SE^ (iii) SCM: 33.9 ± 0.05 ^SE^ (iv) EBST: 36.9 ± 0.05 ^SE^ B. CM (i) USST: BM: 37.5 ± 0.06 ^SE^ Milking: 37.0 ± 0.06 ^SE^ AM: 37.4 ± 0.08 ^SE^ (ii) TSST: BM: 36.9 ± 0.05 ^SE^ AM: 37.0 ± 0.07 ^SE^ (iii) SCM: 34.9 ± 0.06 ^SE^ (iv) EBST: 37.6 ± 0.06 ^SE^	A. SCM Se: 89.0–97.0 Sp: 83.0–94.0 Ac: 96.0–99.0 B. CM Se: 89.0–97.0 Sp: 93.0–98.0 Ac: 98.0–99.0	[86]
54 Sahiwal cows	BM	(i) Teat apex temperature: 34.9 ± 0.11 ^SE^ (ii) TSST: 35.7 ± 0.16 ^SE^ (iii) USST: 37.3 ± 0.09 ^SE^	A. SCM (i) Teat apex temperature: 37.4 ± 0.14 ^SE^ (ii) TSST: 38.6 ± 0.20 ^SE^ (iii) USST: 39.2 ± 0.21 ^SE^ B. CM (i) Teat apex temperature: 37.8 ± 0.07 ^SE^ (ii) TSST: 39.1 ± 0.08 ^SE^ (iii) USST: 40.3 ±0.09 ^SE^	A. SCM Se: 94.0 Sp: 93.0 Ac: 98.0 B. CM Se: 98.0 Sp: 97.0 Ac: 95.0	[87]
BUFFALOES					
35–40 Murrah buffaloes	BM	Seasonal temperature range (i) USST: 30.3–36.8 (ii) TSST: 30.5–36.0	Seasonal temperature range A. SCM (i) USST: 32.5–38.6 (ii) TSST: 32.9–37.6 B. CM (i) USST: 34.3–40.0 (ii) TSST: 34.5–39.1	A. SCM Se: 88.0–98.0 Sp: 85.0–99.0 Ac: 84.0–99.0 B. CM Se: 91.0–97.0 Sp: 89.0–99.0 Ac: 98.0–99.0	[90]
SHEEP					
37 Santa Ines sheep	BM	36.1 (33.6–38.6)	(i) SCM: 36.3 (33.8–39.0) (ii) CM: 35.9 (33.4–38.4)	-	[91]
48 Manchega 35 Lacaune ewes	BM, AM	33.6 ± 0.28 ^SE^ 33.5 ± 1.13 ^SE^	(i) SCM: 33.1 ± 0.28 ^SE^ (ii) CM: 33.5 ± 1.13 ^SE^	-	[92]
GOATS					
104 Skopelos goats	-	(i) Tmax: 38.1 ± 0.58 ^SD^ (ii) Tmean: 36.9 ± 0.58 ^SD^	A. Fibrosis (i) Tmax: 37.8 ± 0.59 ^SD^ (ii) Tmean: 36.5 ± 0.71 ^SD^ B. Fibrosis and asymmetry (i) Tmax: 37.7 ± 0.68 ^SD^ (ii) Tmean: 36.4 ± 0.81 ^SD^	-	[95]

Ac: accuracy, A(M)M: after (morning) milking, A(A)M: after (afternoon) milking, B(M)M: before (morning) milking, B(A)M: before (afternoon) milking, CM: clinical mastitis, CMT: California mastitis test, CV: caudal view, DIMs: days in milk, EBST: eyeball surface temperature, IMI: intramammary infection, L1: level 1, L2: level 2, L3: level 3, LV: lateral view, MG: microbial growth, SD: standard deviation, SE: standard error, Se: sensitivity, Sp: specificity, SCC: somatic cell count, SCM: subclinical mastitis, SME: temperature of short milking tube, Tmax: maximum temperature, Tmean: mean temperature, USST: udder skin surface temperature, TSST: teat skin surface temperature, /2h: every 2 h (additional technical information is summarized in Appendix A).

#### 5.1.4. Goats

In a recent study, Korelidou et al. [95] examined the feasibility of using IRT as a tool for assessing udder health status in 104 Skopelos goats; it was found that the udder halves with both asymmetry and fibrosis had lower mean temperatures compared to the healthy ones (*p* < 0.05, R^2^ = 0.954), underpinning the need to establish appropriate temperature thresholds before the exploitation of IRT as an udder health assessment tool in practice (Table 1 and Figure 4).

### 5.2. Evaluation of Milking Machine Effects on Teats and Udder Skin Surface Temperature

Incorrect practices related to machine milking regarding both operational aspects such as excessive milking vacuum, inappropriate pulsation rate and ratio, liners with wide internal diameter, old and worn-out liners, large mouthpiece chambers, and overmilking can result in mechanical stress, increased temperature, and color changes in the teats post-milking [96]. Tangorra et al. [97] investigated IRT as a tool for estimating the level of stress caused in teats during machine milking in 137 Holstein Friesian cows. They found that the milking process affects the teat skin surface temperature in various degrees, depending on the teat area studied, with maximum temperature performing better in all cases. Alejandro et al. [98] employed IRT as a means for detecting variations in teat tissue temperature associated with milking in Murciano Granadina goats. They noticed that the highest increase in temperature (6.6 °C) occurred at the tip of the teat, while the temperature decreased as the distance from the teat end increased. Mechanical milking increased teat temperature and teat wall thickness due to circulatory changes within the teat wall. In another study, Paulrud et al. [96] used IRT to evaluate the association between soft and standard liner type and overmilking on the teat recovery after milking. The authors observed that the temperature of the teat skin surface steadily declined from the base to the tip (*p* < 0.001) and that the preparatory routine practices before milking, as well as the type of liners, affected these temperatures (e.g., soft liners resulted in lower temperatures). In a recent study by Marnet et al. [99], IRT was evaluated for its capacity to identify changes in teat temperature caused by mechanical milking considering udder symmetry and teat morphology and the detection of udder inflammation. The results indicated that udder symmetry and milking time had no effect on TSST. However, milk yield and parity number had a substantial impact on teat temperature, with older goats displaying a lower drop in temperature after milking. Temperature was reduced at the overall teat level before and after milking; the smallest temperature reduction was observed in the teat barrel (0.4 °C), while the highest was in the teat end (1.1 °C). There was no correlation between IRT values and SCC, indicating that IRT was not effective in detecting mastitis, but it could be useful for adapting milking equipment and improving animal welfare (Table 2).

### 5.3. Setting the Basis for IRT and Evaluation of the Factors Affecting USST

Berry et al. [100] used IRT to examine the daily and intraday fluctuations in USST considering also the environmental effects (Manitoba, Canada), aiming to develop an early mastitis detection protocol based on IRT after adjusting for these fluctuations. According to their findings, USST varied considerably throughout the day due to the circadian rhythm, with a peak observed late in the afternoon and the lowest values being recorded early in the morning, while the lowest variability was observed between 14:00 and 18:00. Additionally, they observed that after a 2 h physical activity (i.e., walking in an outdoor area), USST increased by 1.0 °C, presumably due to the greater exposure to sun radiant heat. When cows did not graze, a rise in USST was observed between 09:00 and 11:00 due to the diurnal increase in the rectal temperature. The authors proposed that previous USSTs in conjunction with ambient temperature could accurately forecast udder temperature. Sathiyabarathi et al. [101] studied body and udder temperature changes according to the lactation stage, milk yield, parity, breed, and season. For this purpose, they used 19 Holstein Friesian crossbred cows and 14 Deoni cows with different productive characteristics; differences attributed to distinct thermoregulatory mechanisms were observed between the two breeds, while USST and body temperature remained constant throughout the study. Deoni cows, characterized by brighter coat color and shorter hair, exhibited greater tolerance to heat stress, resulting in approximately 1.0 °C lower USST compared to crossbred cows. They also reported that USST in the evening was increased by 0.9–1.0 °C in both breeds, possibly due to changes associated with the increased environmental temperatures during daytime and animal activity. Even though the stage of lactation, milk yield, and parity seemed to have no significant effects on USST, the latter was increased by 1.1 °C during the midsummer season. The optimal timing for capturing thermal images with regard to mastitis detection in cows was studied by Yang et al. [102], who concluded that udder thermograms should be taken before milking. In the same study, statistically significant differences in the USST were not observed between the left and the right hind quarters neither before nor after milking, which was attributed to the bilateral symmetry of the similar tissue structure and blood circulation of the udder quarters. However, milk yield was positively correlated with USST after milking, with high-productive cows displaying approximately 1.0 °C higher USST. In their study, Stumpf et al. [103] selected specific areas on the udder for the analysis of the thermograms and examined the correlations among them, as well as with the rectal temperature, aiming to propose a methodology for assessing udder side temperature and developing relevant regression equations. The udder side was selected as a region to evaluate body temperature using the thermal camera and was easily accessible. The studied areas included a horizontal and a vertical rectangle positioned within the image of the udder (ten predetermined spots in the horizontal and vertical rectangle were selected), a vertical and a horizontal line comprising the longest line inside the front quarter, and two diagonal lines traversed from the upper to the lower corner of the udder, one from the right to the left and vice versa. All areas demonstrated their capacity to efficiently evaluate the udder temperature and predict the rectal temperature. The maximum temperature attained by deploying a horizontal triangle within the udder’s lateral side was the optimal method (R^2^ = 0.816). In the study by Korelidou et al. [104], 104 and 236 Skopelos goats were prospectively (study A) and cross-sectionally (study B) studied, respectively, in order to estimate the effects of body condition score (BCS) on the USST, TSST, and udder cleft surface temperature (UCST). Their findings indicated that a one-degree decrease in BCS was associated with an increase of (i) 0.7 °C and 0.8 °C in the max UCST, (ii) 0.9 °C and 1.3 °C in the mean UCST, (iii) 0.3 °C and 0.5 °C in the max USST, and (iv) 0.6 °C and 0.7 °C in the mean USST for studies A and B, respectively (Table 3).

### 5.4. Integration of Thermal Imaging Data into Prediction Algorithms

Watz et al. [105] carried out a study to compare automatic image analysis software with the conventional method (manual analysis). For this purpose, they intramammarily infused *E. coli* to induce clinical mastitis in five healthy Holstein Friesian cows. The manually obtained temperatures were higher in comparison to the automatic method that omitted warmer regions during segmentation. The average USSTs obtained with both methods were strongly correlated, and the maximum temperature displayed better diagnostic performance. In a more recent study, Khakimov et al. [106] developed an algorithm for the evaluation of udder health status using IRT. For this purpose, they enrolled 250 Yaroslav cows and recorded their milk yield and USST. They found that USST was differentiated according to the udder health status; namely, it ranged from 32.0 to 35.9 °C for healthy cows, from 36.0 to 38.3 °C for cows with subclinical mastitis, and from 38.4 to 39.0 °C for cows with clinical mastitis. They also found that USST was significantly correlated with milk yield only in cows with clinical mastitis (R^2^ = 0.886). In particular, when USST exceeded 36.0 °C, milk yield started to decline. The diagnostic capacity of IRT was also evaluated by Wang et al. [107], who applied deep learning technologies and relied on distinguishing the differences between the left and right USST, as well as between USST and ocular temperature to detect mastitis in cows (SCC > 200,000 cells/mL). The authors acknowledged the occurrence of misdetections when assessing udder health statuses by IRT, primarily attributed to external factors (e.g., dirt) affecting USST, as well as to animals’ physiological parameters (e.g., milk yield, parity, lactation period, etc.). The method was promising and achieved an accuracy, specificity, and sensitivity of 87.6%, 84.6%, and 96.3%, respectively. The lower accuracy of mastitis detection based on deep learning compared to the manual labeling was attributed to errors in animal positioning, leading the authors to the conclusion that although the method is promising and can be integrated into animal monitoring systems, the use of a depth camera could further enhance accuracy. In a similar study, Xudong et al. [108] developed an algorithm for the detection of mastitis in dairy cows based on eye temperature and USST. The algorithm included techniques both for the improvement of image analysis and for the enhancement of the target detection. The algorithm’s accuracy, sensitivity, and specificity were 83.3%, 92.3%, and 76.5%, respectively, with the authors suggesting that the algorithm was effective in identifying mastitis cases (SCC > 200,000 cells/mL). Likewise, Chu et al. [109] worked on developing a computer vision-based model using deep learning to automatically detect mastitis in cows. For their study, they captured thermal videos from 196 Holstein Friesian cows and combined udder temperature and udder size features. In particular, they employed CenterNet deep learning to identify specific key points on the udder, facilitating the automatic calculation of its size and the evaluation of various udder temperature features that included correlations between the two udder sides and eye temperatures; their proposed method achieved an accuracy of 88.6%, while the sensitivity and specificity for the detection of mastitis were 87.5% and 94.0% for clinical mastitis and 81.3% and 91.9% for subclinical mastitis (SCC > 200,000 cells/mL), respectively. Despite the promising results, the authors stated that further work is necessary to improve sensitivity and prevent underdiagnosis, underlining also the need for optimization of the classification methodology. In another study, a deep learning Convolutional Neural Network (CNN) model based on udder thermograms to automatically detect subclinical and clinical mastitis in Murrah buffaloes was developed by Gayathri et al. [110]. This study found a strong correlation between USST and CMT and log_10_SCC (r = 0.87 and r = 0.88, respectively, *p* < 0.01); buffalos with clinical (CMT = 3 and log_10_SCC > 5.48) and subclinical (CMT = 1 or 2 and log_10_SCC 5.00–5.48) mastitis had higher mean USST by 1.3 °C and 2.6 °C, respectively, compared to healthy animals (CMT = 0 and log_10_SCC < 5.0). The trained model demonstrated high sensitivity (95.2% and 96.0%, respectively) and specificity (91.2% and 93.5%, respectively) (Table 4). 

## 6. Future Perspectives, Challenges, and Limitations of Infrared Thermography for the Diagnosis of Mastitis

### 6.1. Perspectives

Information and communication technology along with the use of smart network objects, remote monitoring, modern data collection through sensors, rapid data transfer, and vast data storage through the Internet of Things (IoT) have resulted in the modernization of agriculture and the advancements in precision livestock farming (PLF) [111]. Remote and wearable sensors combined with algorithms can be applied for the assessment of body weight and body condition score, detection of lameness and mastitis, monitoring feed intake and behavior, quantification of pain and stress, and positioning systems [112,113]. Precision livestock farming technologies aim to create a management system for the ongoing and real-time monitoring of health, welfare, productivity, and environmental impact [114,115]. In this context, IRT cameras can work synergistically with modern machine learning models to extract thermal details and assess animal’s health statuses [63]. These automated systems can enable the acquisition and processing of large amounts of data enhancing efficiency, reducing manpower, and improving animal production practices [116].

Early detection of mastitis can contribute to mitigating losses associated with degraded milk yield and quality [117]. Infrared thermography is a promising tool for improving udder health status in dairy animals [84] due to its low cost and capacity to provide rapid and precise udder temperature measurements [37]. In addition, the technology is safe and user friendly and can be incorporated into routine clinical examinations as an auxiliary *in situ* diagnostic tool, given its capacity to minimize animal handling and restrain demands, thereby reducing stress on the animals [80].

Milk temperature can possibly serve as an indicator of udder health [118] and may be useful in the diagnosis of mastitis [119]. Currently, sensors have been incorporated into milking machines for the measurement of milk temperature; yet, the existing literature on the use of IRT for determining milk temperature is scarce. This has been recognized by Gayathri et al. [86,120], who tested the capture of thermal images of short milking tubes, integrated into a machine milking system, for the measurement of milk temperature. Their findings indicated that milk temperatures measured by IRT in Murrah buffaloes and Sahiwal cows with mastitis were higher (34.9 °C in both animals) compared to the milk temperatures of animals with subclinical mastitis (33.4 °C and 33.9 °C, respectively) and the healthy ones (31.2 °C and 32.8 °C, respectively), verifying the potential application of IRT in milk as an indicator of mastitis. However, further research is necessary to elucidate the relationship between milk temperature, thermogram readings, and mastitis occurrence.

### 6.2. Challenges

However, IRT’s regular application is challenged by factors related to the surrounding environment, animal physiology, health and welfare status, and the operator’s capability to determine and follow standard procedures when acquiring and analyzing thermal images (Table 5) [62].

Infrared thermography is affected by environmental conditions; namely, wind, rain, and ambient temperature, which can modify animals’ udder skin temperature due to evaporation and blood flow changes to the skin during thermoregulation [85]. Additionally, elevated ambient temperature and humidity levels can increase peripheral vasodilation and body temperature and potentially result in inconsistencies in temperatures obtained by thermal images [84], while exposure to sunlight or high ambient temperature can increase reflected radiation and thus surface temperature, leading to inaccurate measurements [121]. In general, meteorological conditions affect USST, with healthy animals displaying different USST temperatures across seasons [90,101,104]. In addition, under field conditions, animals are rarely clean and dry [72], and the presence of urine, manure, dirt, or foreign bodies on udder skin impacts emissivity, leading to variations in USST and thus affecting the quality of thermograms and readings thereof [85]. Dirt removal from the udder through rubbing may increase temperature, while washing and cooling may decrease it [77]. The aforementioned facts necessitate the capturing of thermal images under real-world conditions and various scenarios so that all the affecting factors can be quantified. Animal-related factors, such as physical activity and stress, as well as the effects of the circadian rhythm, production level, and time of feeding and milking, have also been found to affect USST [81]; therefore, it is critical to consider these factors when determining the most appropriate time for capturing thermograms. Additionally, USST is affected by the extent of hair coverage in the udder [72]. Particularly, the type, color, and thickness of hair affect the skin’s capacity to absorb heat [122]; darker hair absorbs more radiation, while thicker hair provides higher insulation levels blocking the energy from being reflected and detected by the thermal cameras [62]. At the same time, differences in udder appearance are evident both between and within breeds, while the breed itself [92,101], age [99], milk yield [102], and body condition score [99] have been found to affect USST; thus, appropriate protocols need to be developed considering phenotypic differences with regard to traits affecting USST temperature.

When capturing thermal images from the caudal side of the udder, the animals should be restrained, their tail raised and legs kept apart to ensure a clear view of the udder; therefore, it is crucial to decide where to stand or install the cameras so that the udder is fully visible without needing to intervene on the animals and simultaneously protecting the cameras from dust, mud, and the animals themselves. Another challenge encountered during IRT’s application is setting the anatomical boundaries of the udder since the inside part of the legs has the same temperature as the skin of the udder [77]. Currently, manual analysis of images is time consuming and subjective; therefore, it is likely to compromise the accuracy of the results [77]. Optimization of the thermogram analyzing process, standardization of the area to be measured and analyzed using the software, and the automatic outlining of the udder through the development of appropriate segmentation algorithms could contribute to determining a standardized procedure [107,123]. Factors including emissivity, the reflected temperature, the angle of capturing, and the distance between the camera and the object should be taken into consideration during this process to achieve high accuracy levels [124]. Finally, cost is another significant challenge encompassing the type of camera that will be purchased; there are several thermal cameras in the market with varying settings, different sensitivities and resolutions, and a price range between EUR 300 and 45,000. Currently, low-cost portable thermal cameras that can be attached to smartphones are commercially available; however, their effectiveness on animals’ body temperature recording has not been evaluated yet. Additional costs include purchasing the software for the analysis of the thermogram, as well as the training cost of the end-users to handle the equipment accurately and interpret the results correctly.

### 6.3. Limitations

Detection of mastitis with IRT has some further limitations related to the inherent nature of the disease. In particular, subclinical mastitis lacks visible signs of udder inflammation or other systemic effects [125,126], making its detection using IRT even more challenging. Thus, targeted studies are needed to understand how udder temperature changes as subclinical mastitis progresses.

Intramammary infections can be caused by many different pathogens, resulting in variations in the severity of the subclinical mastitis and the degree of lesions and inflammation. Moreover, the severity of subclinical mastitis relies on the host’s genetic resistance to mastitis and response to mastitis-causing pathogens, which are dependent on the breed, and immunological status of the animal [127,128], as well as the virulence of pathogens [129]. The aforementioned factors urge the investigation of the effects of various subclinical mastitis-causing pathogens on USST in different breeds.

Furthermore, subclinical mastitis can be localized within specific parts of the udder, displaying non-usual patterns that may limit the accuracy of IRT for early diagnosis, depending on the capturing site. Also, inflammatory processes localized in deeper udder tissues may not generate enough surface heat to reach the skin’s surface or alter blood flow; thus, they may not be detectable by IRT [78], which typically characterizes bacterial stains that result in persistent infections [130].

Different stages of subclinical mastitis are associated with the diagnostic capacity of IRT. In particular, pathogens’ invasion and intramammary infections thereof trigger the immunological and inflammatory responses at the initial stages [131], while at later stages, subclinical mastitis is mainly characterized by repair and less by inflammation processes, leading to the underdiagnosis of chronic cases of subclinical mastitis when IRT is used [132].

## 7. Conclusions

Implementing automated systems for remote temperature measurement, such as IRT, can contribute to decreasing total farm expenses by facilitating faster disease detection, prompt medication administration, and thus improved animal health and welfare. The existing literature on the efficacy of IRT in evaluating udder health status is particularly focused on dairy cows with contradictory results. This variation is attributed to the differences in camera settings, environmental conditions, and definitions of udder health status across studies. Factors related to the subject being observed, the environment, and the process of capturing and analyzing the image can impact the quality of thermograms. Therefore, it is crucial to detect and regulate these parameters when implementing IRT. Research efforts in the field of thermal imaging for the assessment of udder health should be focused on naturally occurring mastitis cases under real-world, field conditions to adjust for environmental and animal-related factors (breed, age, parity, milk yield, milking practices, physiological factors) that influence udder temperature and the diagnostic performance of IRT. Among them, the challenges associated with subclinical mastitis (lack of visible alterations, pathogen-related changes in temperature patterns, stage of infection) need to be considered during the design of relevant studies. In any case, IRT can serve as a non-invasive, rapid screening tool that has the potential to be integrated into advanced precise livestock farming technologies for continuous animal monitoring; however, standardized methods for capturing IRT images (e.g., the distance, the region of interest, and the viewpoint), the most appropriate udder anatomical regions, and threshold values of normal udder temperature values have yet to be established. The integration of algorithms and deep learning image analysis into thermal cameras could solve some of these challenging issues and facilitate the automation of IRT for its widespread use.

## Figures and Tables

**Figure 1 animals-14-02691-f001:**
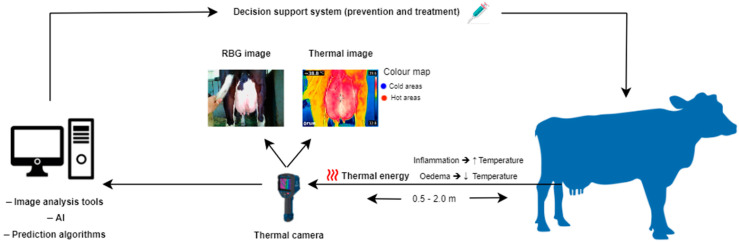
Illustration of thermal imaging for the management of cow’s udder health.

**Figure 2 animals-14-02691-f002:**
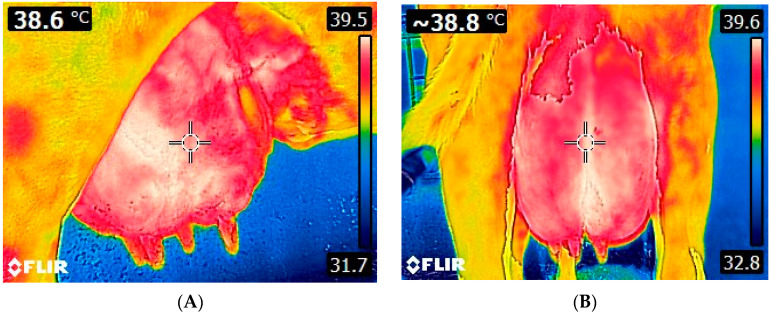
Thermal images of a cow udder. Lateral (**A**) and caudal (**B**) view of the udder. Images were taken using FLIR E8-XT (FLIR Systems Inc., Wilsonville, Oregon, USA), which displays a 320 × 240 resolution (source: Laboratory of Anatomy and Physiology of Farm Animals, Agricultural University of Athens).

**Figure 3 animals-14-02691-f003:**
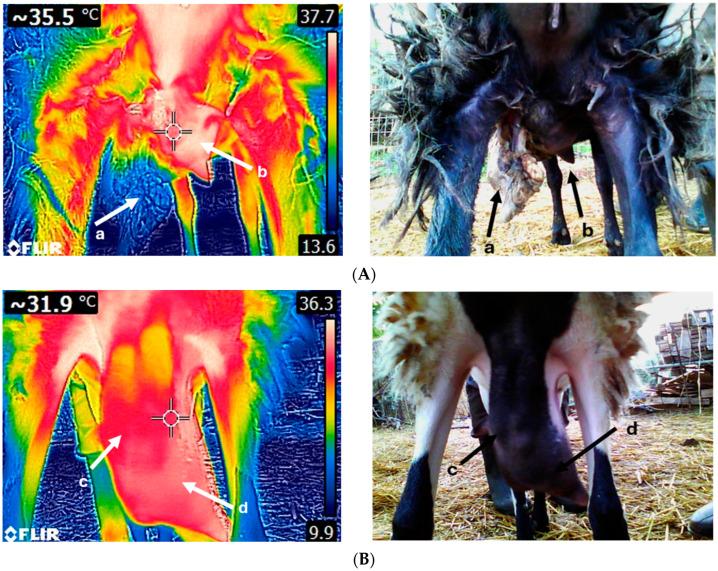
Thermal image of a sheep suffering from gangrenous mastitis (**A**). The left udder half (a) has been necrotized, while the right udder half (b) displays an increased temperature. Thermogram of a sheep udder with chronic mastitis (**B**); both udder halves (c,d) are fibrotic. Udders with acute clinical mastitis exhibit higher temperatures compared to chronic cases. Images A and B were taken on the same day and time using FLIR E8-XT (FLIR Systems Inc., Wilsonville, Oregon, USA) at a 320 × 240 resolution (source: Laboratory of Anatomy and Physiology of Farm Animals, Agricultural University of Athens).

**Figure 4 animals-14-02691-f004:**
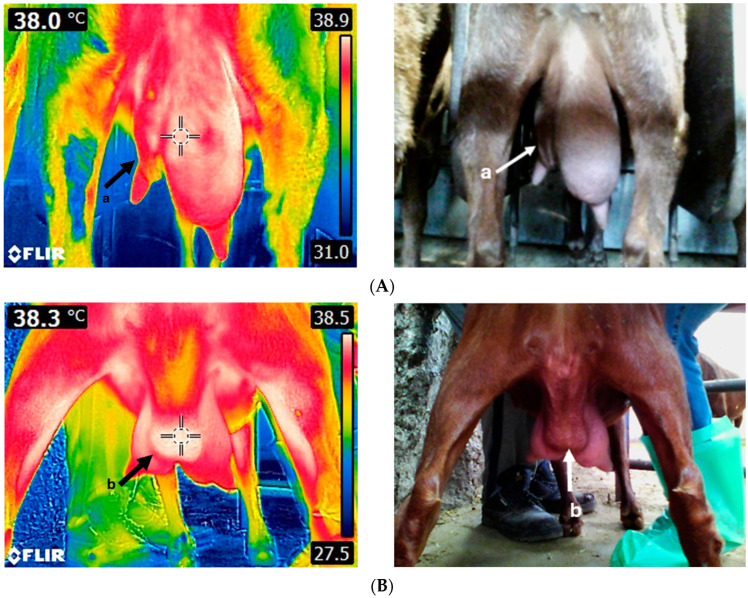
Thermal image of an unhealthy goat udder (**A**). The left udder half is severely asymmetric and fibrotic (a). Thermal image of a goat udder with an abscess (**B**). The abscess (b) is located on the udder cleft and displays a higher temperature compared to the rest of the udder. Images were taken using FLIR E8-XT (FLIR Systems Inc., Wilsonville, Oregon, USA), which displays a 320 × 240 resolution (source: Laboratory of Anatomy and Physiology of Farm Animals, Agricultural University of Athens).

**Table 2 animals-14-02691-t002:** Udder and teat skin surface temperature pre- and post-milking.

Animals	Temperature (°C) (Mean ± SD/Mean ± SE)		Reference
	Before Milking	After Milking	
8 Danish Holstein cows	A. Extended liner: (i) Not overmilked: 34.4 (ii) Overmilked: 33.8 B. Soft liner: (i) Not overmilked: 33.8 (ii) Overmilked: 33.8	A. Extended liner: (i) Not Overmilked: 35.3 (ii) Overmilked: 35.6 B. Soft liner: (i) Not overmilked: 34.6 (ii) Overmilked: 34.9	[96]
30 Murciano- Granadina goats	Area (distance from teat end) (i) 1 cm: 28.4 ± 0.26 (ii) 2 cm: 31.5 ± 0.26 (iii) 3 cm: 33.2 ± 0.26 (iv) udder: 34.3 ± 0.26	Area (distance from teat end) (i) 1 cm: 33.3 ± 0.26 (ii) 2 cm: 34.1 ± 0.26 (iii) 3 cm: 34.6 ± 0.26 (iv) udder: 35.9 ± 0.26	[98]
137 Holstein Friesien cows	-	Purple teat color (i) Teat base Tavg: 34.5 ± 0.09 ^SE^, Tmax: 35.1 ± 0.10 ^SE^ (ii) Teat centre Tavg: 35.1 ± 0.08 ^SE^, Tmax: 36.1 ± 0.08 ^SE^ (iii) Teat tip Tavg: 35.2 ± 0.14 ^SE^, Tmax: 37.0 ± 0.10 ^SE^	[97]

Ac: accuracy, A(M)M: after (morning) milking, A(A)M: after (afternoon) milking, B(M)M: before (morning) milking, B(A)M: before (afternoon) milking, CV: caudal view, LV: lateral view, Se: sensitivity, Sp: specificity, Tmax: maximum temperature, Tavg: average temperature (for additional technical info, see Appendix A).

**Table 3 animals-14-02691-t003:** Studies on physiological factors affecting thermal camera readings.

Animals	Stage of Milking Period (DIM) (Mean ± SD)	Milk Yield (Mean ± SD)	Anatomical Region of the Udder	Time	Temperature (°C) (Mean ± SD)	Reference
10 Holstein Friesian cows	104	-	CV	(i) Study 1: 30 min before exercise and immediately after returning (ii) Study 2: /2h	(i) Before exercise: 33.4 (ii) After exercise: 34.5	[100]
102 Holstein Friesian cows	76 ± 67 ^SD^	13.5 ± 4.7 ^SD^ kg	CV	BM, AM	A. Milking/quarter (i) Left: BM: 35.9 ± 0.87 ^SD^, AM: 37.2 ± 0.83 ^SD^ (ii) Right: BM: 35.8 ± 0.93 ^SD^, AM: 37.1 ± 0.85 ^SD^ B. Milk yield (i) <10 kg: BM: 35.4 ± 0.21 ^SD^, AM: 36.3 ± 0.20 ^SD^ (ii) 10–15 kg: BM: 35.8 ± 0.11 ^SD^, AM: 37.2 ± 0.12 ^SD^ (iii) > 15 kg: BM: 36.1 ± 0.12 ^SD^, AM: 37.3 ± 0.10 ^SD^	[102]
19 Holstein and 19 Girolando cows	(i) Holstein: 249 ± 68 ^SD^ (ii) Girolando: 136 ± 97 ^SD^	(i) Holstein: 14.8 ± 2.6 ^SD^ L (ii) Girolando: 13.6 ± 4.89 ^SD^ L	LV	Between morning and afternoon milking	-	[103]
19 crossbred Holstein Friesian x Bos indicus and 14 Deoni cows	-	(i) Crossbred: 14.4 ± 0.2 ^SD^ kg (ii) Deoni: 3.5 ± 0.1 ^SD^ kg	LV, CV	BMM, BAM	A. Milking time (i) Morning Crossbred: 37.2 ± 0.03 ^SD^, Deoni: 36.2 ± 0.07 ^SD^ (ii) Evening Crossbred: 38.2 ± 0.06 ^SD^, Deoni: 37.2 ± 0.08 ^SD^ B. Stage of lactation and milk yield Early, mid, late, low, and high milk yield: Crossbred: 37.2 ± 0.01 ^SD^, Deoni: 36.2 ± 0.01 ^SD^ C. Season—Crossbred (i) Spring: 37.2 ± 0.01 ^SD^ (ii) Winter: 36.4 ± 0.01 ^SD^ (iii) Summer: 37.8 ± 0.01 ^SD^	[101]
Study 1: 104 Skopelos goats Study 2: 236 Skopelos goats	Study 1: whole lactation period Study 2: mid-lactation	-	CV (udder, teats, udder cleft)	AM	Study 1: (i) TSST: max: 32.9–37.3, mean: 31.9–36.9 (ii) UCST: max: 36.3–38.0, mean: 35.1–37.4 (iii) USST: max: 37.1–38.4, mean: 35.2–37.5 Study 2: (i) TSST: max: 34.2 ± 0.99 ^SD^, mean: 33.0 ± 0.95 ^SD^ (ii) UCST: max: 36.8 ± 0.88 ^SD^, mean: 35.2 ± 1.28 ^SD^ (iii) USST max: 37.6 ± 0.68 ^SD^, mean: 35.6 ± 0.87 ^SD^	[104]

AM: after milking, B(M)M: before (morning) milking, B(A)M: before (afternoon) milking, CV: caudal view, LV: lateral view, Max: maximum, SD: standard deviation, UCST: udder cleft surface temperature, USST: udder skin surface temperature, TSST: teat skin surface temperature, /2h: every 2 h (for additional technical info, see Appendix A).

**Table 4 animals-14-02691-t004:** Studies where thermal imaging was integrated into prediction algorithms.

Animals	Stage of Milking Period	Anatomical Region of the Udder	Time	Temperature (°C) (Mean ± SD/Mean ± SE)	Sensitivity/Specificity/ Accuracy (%)	Reference
5 Holstein Friesien cows	-	CV	/2h before and after *E. coli* infusion	(i) Before *E. coli* challenge Automatic: 36.3, Manual: 37.3 (ii) After *E. coli* challenge Automatic: 38.7, Manual: 39.6	(i) Automatic Se: 93.8, Sp: 95.0 (ii) Manual Se: 93.8, Sp: 96.4	[105]
30 Holstein Friesian cows	Middle of the lactation period	-	BM	SCM: USST by 0.8 °C higher than OST	Se: 92.3, Sp: 76.5, Ac: 83.3	[108]
250 Yaroslav cows	5th/6th month of lactation	CV, LV	BM, AM	(i) Healthy: 32.0–35.9 (ii) SCM: 36.0–38.3 (iii) CM: 38.4–39.0	-	[106]
105 Holstein Friesian cows	-	LV	BM	0.72 °C difference in USST between mastitic and healthy USST > OST (by 0.8 °C)	Se: 96.3, Sp: 84.6, Ac: 87.6	[107]
196 Holstein Friesian cows	Middle of lactation	LV	BM	-	(i) SCM: Se: 81.3, Sp: 91.9, Ac: 88.6 (ii) CM: Se: 87.5, Sp: 94.0, Ac: 88.6	[109]
40 Murrah buffaloes	Various	LV	BM	(i) Healthy: 34.5 ± 0.04 ^SE^ (ii) SCM: 35.8 ± 0.03 ^SE^ (iii) CM: 37.1 ± 0.07 ^SE^	(i) SCM: Se: 95.2, Sp: 91.2 (ii) CM: Se: 96.0, Sp: 93.5	[110]

Ac: accuracy, AM: after milking, BM: before milking, CM: clinical mastitis, CMT: California mastitis test, CV: caudal view, LV: lateral view, OST: ocular surface temperature, Se: sensitivity, SCM: subclinical mastitis, Sp: specificity, USST: udder skin surface temperature, /2h: every 2 h (for additional technical info, see Appendix A).

**Table 5 animals-14-02691-t005:** Summary of the factors affecting udder thermal imaging performance.

Animal	Environment	Technology Aspect
Species Breed Udder morphology Age Milk yield Body condition score Udder-related diseases Stress Physical activity Circadian rhythm	Season of the year Climatic conditions (wind, sunlight, rain, humidity) Time of day Feeding pattern Dusty environment Filthy udders (mud, urine, foreign bodies)	Type of camera Personnel training Settings of the camera (e.g., emissivity, distance, angle from the animal) Thermogram analysis method—software

References [62,72,77,81,84,85,90,92,99,101,102,104].

## Data Availability

Not applicable.

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
