# Peer review of "Infrared Thermography as a Diagnostic Tool for the Assessment of Mastitis in Dairy Ruminants"

_animals, 2024, doi:10.3390/ani14182691_

Round 1

Reviewer 1 Report

Comments and Suggestions for Authors

Some comments and corrections:

-Very nice and informative work and thanks for the focus. But, needs to be more focused on the main issues of the study..... It is better to more focused on the main topical point(s) and word(s). [your title of study is: “Infrared thermography as a diagnostic tool for the assessment of mastitis in dairy ruminants”... please stick on this topical points throughout, and do not go far beyond that area.... reader might not want to read such unnecessary ones...

-Briefly address something on the IRT application in dairy and vet sciences, worldwide and where we are now? May be in the introduction.

-Please clearly add on the issues of challenging application and limitations of IRT in mastitis detection; although some parts is written but should be clearly highlighted and readers might want to see such.

-It is also difficult to grasp the meaning of saying in the first reading... and should be simplified with more focused and illustrations/schemes to show/highlight the notions. Please elaborate in the revise version some unnecessary ones should be removed throughout the text.

-Challenging and limitation points of IRT-based subclinical mastitis diagnosis (especially early diagnosis) remained unnoticed. Please add specific subtitle and written text in the MS.

-Line 67, The last decade, several studies have been conducted on the application of IRT in.... please remove the “The last decade,” here.

-Line 71,..... please replace “use” for the “utilization” of IRT... also elaborate other place of the text throughout (there more… eg, lines 145, 177, 477 ….). Please revise and correct.

-Lines 80-119, unnecessary and should be removes and just written as 2 sentences (if).....

-Line 132, ….efficient diagnosis, prevention and treatment….

-About the notion of diagnostic values of IRT (sensitivity 95.6% and specificity 93.6%), are you sure and with what circumstances? for which kind of mastitis? Please clearly elaborate.

-What is the main message of Fig. 1 in relation to the mastitis diagnosis? Please elaborate. Also other figs’ captions should be very precisely and clearly addressed.

-What is the main message of Table 1 (The table(s) is user unfriendly and hard to grasp) in relation to the mastitis diagnosis? Please also highlight those main ones; please elaborate.

Line 303, ....Ec > 50 units? Units?

Line 465,..... and narrowing of the teat canal. And the outcome related to IRT-based temperature? Please add here.

Line 488, ... a rise in USST was observed.... how much?

-For table 6 I would add some references?

Line 648, .... 300 and 45.000 euros.... what is “.” Here [45.000]?

-Milking (pre-, post and during) vs IRT, and please make a very informative table/fig/scheme?

-Temperature on the surface of udder skin, inside the ductal lumen and milk? Should be added and some also from your own?

-IRT is yet to be an effective tool to diagnose the occurrence of mastitis  .... especially the sub-clinical one... this should be somehow added in the conclusion part as well.

Good luck

Comments on the Quality of English Language

needed some.

Reviewer 2 Report

Comments and Suggestions for Authors

The aim of this review paper was to present the current state of knowledge regarding the detection of mastitis in dairy ruminants using infrared thermography. The summary of the different areas was clear and concise, with sufficient and up-to-date information. Furthermore, the strengths, weaknesses, opportunities, and challenges of infrared thermography were addressed objectively based on the evidence available to date. The manuscript is well written and has practical implications for dairy producers.

Below are suggestions and comments on various aspects that the authors need to address before the paper can be published.

Line 48: The reference only pertains to New Zealand.

Line 51: "Apparent changes in the milk." The changes in milk composition require analytical tests; they are not apparent changes.

Lines 52-56: These tests are used for the diagnosis of subclinical mastitis. Please clarify this point.

Line 77: Considering the second objective, point i), of your manuscript: It is unclear how the adoption of IRT at the herd level for mastitis surveillance was emphasized.

Lines 81 and 99: I suggest referring to mastitis in general, not just clinical mastitis.

Lines 128-129: This sentence may be confusing. I suggest rewriting it as: "Coagulase-negative staphylococci have been primarily isolated in subclinical mastitis cases."

Lines 141-143: Include bibliographic references for regulated thresholds for somatic cell count in the USA.

Lines 168-198: Additionally, a disadvantage of PCR is the inability to identify viable pathogens.

Lines 253-257: I suggest describing the findings regarding the ability of IRT to detect subclinical mastitis using 200,000 cells/mL as a cut-off, as this is a threshold more commonly used in the dairy industry.

Lines 331 and 334: The abbreviation "SMT" has not been previously defined, which makes it difficult to understand. Please define it when first introduced.

Line 486: Clarify the region or country where the measurements were taken.

Line 487: Clarify what is meant by "2-hours exercise."

Line 500: The environmental temperature is usually lower in the evening.

Lines 304, 497, 509: What do you mean by "ca."? To make the text more reader-friendly, I suggest using "approximately."

Line 576: The abbreviation "CNN" has not been previously defined. Please define it when first introduced.

Lines 61, 84 ([12]), 86, 109, 112, 126, 132-137, 589, 605, 636: The references used do not seem to be the most appropriate. Consider revising these references.

Table 1, column named "Author" (reference [79]): "1st lactation" is not a stage of the milking period. Please revise.

Table 1, column named "Milk Yield": Not all milk production values are expressed in kilograms. Please clarify the units used.

Table 1, column named "Time": What does "/2h" mean? Clarify this notation.

Table 1, for GOATS, column named "Temperature": The dispersion measures (e.g., standard deviation, standard error) are missing. Please include them.

Table 2, columns named "Before" and "After Milking": The dispersion measures (e.g., standard deviation, standard error) are missing. Please include them.

Tables 1, 2, and 4, columns named "Sensitivity/Specificity": Add "Accuracy" to these columns.

Tables 3 and 4: The format of Tables 1 and 2 is not consistent with the "Animals" column. Please ensure uniform formatting across all tables.

Reviewer 3 Report

Comments and Suggestions for Authors

General Comments

This is a good review. The manuscript is well written, well organized and well referenced. The manuscript appears to be free of spelling, grammar, and typo errors. The tables and figures are appropriate and illustrative.  Subject to the minor revisions identified below I would recommend publication. The authors are to be commended for their efforts.

Minor Comments:

1.     The review is missing some of the pertinent early work regarding the use of infrared detection of mastitis in both humans and cattle. The authors could perhaps at least suggest their manuscript recognizes that earlier work was performed but their manuscript focuses on research within the past 15 - 20 years?

2.     The manuscript would benefit from a discussion of where this application of technology is moving towards. To be applicable to and have utility in  the scale of modern agriculture the thermal technology needs to become an automated precision farming tool rather than the domain of hand held detectors in primarily R and D facilities. To this end there have been some significant achievements  in precision farming incorporating computerized, internet enabled and non invasive detection systems. A reference for example to the Burleigh Dodds Publication edited by D. Berckmans. 2022. Advances in precision livestock farming, would I believe be appropriate and additive for the manuscript.

3.     L266 – makes reference to the work by Hovinen et al regarding an ecoli mastitis induction model. This was not the first use of an induction model in this regard ( Scott, S.L. et al. 2000. Use of infrared thermography for early detection of mastitis in cows. Proc. Of Agric. Inst. Of Canada. Agri-Food. 2000. Winnipeg).

4.     For Figures 1, 2, 3 It would be useful for a reader if the camera make, model and image resolution were added.

5.     For Table 1. This is a large amount of information for one table. The authors might consider breaking this into several components. Perhaps by species?

Round 2

Reviewer 1 Report

Comments and Suggestions for Authors

I see the authors have done and revised their nice topic work well and after small revision, it should be acceptable.

Some:

-About the “allergies vs mastitis” if you have specific reference would be useful to add? About reference #12 for allergies might not be relevance?

-On tables, for addressing the number etc.. I would add the abbreviations (like SD or SE) in is related caption. Please elaborate.

-About the notion of diagnostic values of IRT (sensitivity 95.6% and specificity 93.6%), this should be addressed with some previous studies (if you want to add the exact numbering)? Please precisely elaborate; or write as “acceptable range”... with reference (if any)?

-Regarding actual milk temperature, if IRT give exact numbering of it? Please challenging add wording in the conclusion as (maybe limitation and needs to be further studies...)?

Very good luck

Comments on the Quality of English Language

-
